# Both *Saccharomyces boulardii* and Its Postbiotics Alleviate Dextran Sulfate Sodium-Induced Colitis in Mice, Association with Modulating Inflammation and Intestinal Microbiota

**DOI:** 10.3390/nu15061484

**Published:** 2023-03-20

**Authors:** Xinge Xu, Jingwei Wu, Yuxin Jin, Kunlun Huang, Yuanyuan Zhang, Zhihong Liang

**Affiliations:** 1College of Food Science and Nutritional Engineering, China Agricultural University, Beijing 100083, China; 2Beijing Laboratory for Food Quality and Safety, College of Food Science and Nutritional Engineering, China Agricultural University, Beijing 100083, China; 3Beijing Key Laboratory of Zoo Captive Wildlife Technology, Beijing 100044, China

**Keywords:** *Saccharomyces boulardii*, postbiotics, ulcerative colitis, inflammatory, intestinal microbiota, intestinal barrier, dextran sulfate sodium (DSS)

## Abstract

Objective: To investigate the effect of *Saccharomyces boulardii* and its freeze-dried and spray-dried postbiotics on the intervention and potential mechanism of dextran sulfate sodium (DSS)-induced ulcerative colitis in mice. [Methods] After the acclimation period of C67BL/6J mice, a colitis model was constructed by applying 2% DSS for 7 d, followed by 7 d of intervention. Subsequently, the disease activity index (DAI), organ index, colon length, colon HE staining of pathological sections, ELISA for blood inflammatory factors (Interleukin (IL)-1β, IL-6, IL-10, Tumor necrosis factor (TNF)-α), Real time quantitative polymerase chain reaction (RT-qPCR) to determine the levels of colonic inflammatory factors (IL-1β, IL-6, IL-10, TNF-α), Occludin gene expression, and intestinal flora were assessed to evaluate the protective effects of *S. boulardii* and its postbiotics on colitis in mice. Results: Compared with the DSS group, *S. boulardii* and the postbiotics interventions effectively improved colonic shortening and tissue damage, increased the expression of intestinal tight junction protein, reduced the secretion of pro-inflammatory factors, increased the secretion of anti-inflammatory factors, and maintained the homeostasis of intestinal microorganisms. Postbiotics intervention is better than probiotics. Conclusions: *S. boulardii* and its postbiotics can effectively alleviate DSS-induced colitis in mice through modulating host immunity and maintaining intestinal homeostasis. Postbiotics are promising next-generation biotherapeutics for ulcerative colitis treatment.

## 1. Introduction

Inflammatory bowel diseases (IBD) are a group of chronic intestinal inflammatory diseases that mainly include ulcerative colitis (UC) and Crohn’s disease (CD) [1]. UC is a chronic inflammatory disease affecting the colon and rectum that mostly manifests as abdominal pain, diarrhea, and rectal bleeding [2]. UC has become a global disease in recent years, and its prevalence and incidence are still on the rise, especially in the rapidly developing newly industrialized countries, posing a serious threat to people’s quality of life and health [3]. A combination of environmental factors, genetic susceptibility, epithelial barrier defects, and dysregulated immune response contribute to the development and progression of UC [4]. The gut microbiota, as a major environmental driver, profoundly influence the immune composition of the host under physiological and pathological conditions, and disturbances in the gut microbiota lead to immunological dysregulation, which may underlie diseases such as inflammatory bowel disease [5]. It has been noted that although some medical treatments such as steroids, immunomodulators, and antibodies are available for UC patients, there are problems of low treatment response and high incidence of recurrences, thus UC treatment imposes a considerable financial burden on the healthcare system, highlighting the need for novel therapeutics to improve disease management [6,7]. Studies have shown that the use of probiotics can reduce the symptoms associated with UC and that their effects may be related to the regulation of the gut microbiota, thus attracting much attention [8].

*Saccharomyces boulardii* is a type of *S. cerevisiae*, a nonpathogenic and probiotic yeast, which belongs to facultative anaerobic fungi and exerts anti-cancer, immune modulation, antibacterial, antiviral, and antioxidant functions in the body [9]. It was found to be the most abundant fungal genus in healthy individuals [10]. *S. boulardii* has been used as a probiotic since the 1950s for prevention and treatment of antibiotic-associated diarrhea and has demonstrated efficacy in pilot studies in patients with inflammatory bowel disease (IBD) [11]. However, it should not be overlooked that the exact mechanism by which *S. boulardii* relieves the symptoms of colitis is not fully understood. Furthermore, some safety concerns have been raised about the use of this probiotic, with a small number of studies finding fungemia [12]. It has also been shown that non-curative *Escherichia coli* will cause damage to the wall of *S. boulardii*, affecting its ability to exert a direct probiotic effect on pathogenic *E. coli*. As a result of that, the intestinal microbiota contains high levels of non-curative *E. coli*, which will affect the efficiency of *S. boulardii* in the intestine [13]. All this evidence questions the safety of this preventative biotherapy in clinical use, so it would be of interest to develop products that not only resemble the functional repertoire of probiotics but also pose no or the near absence of risk to users.

Postbiotics are a new mean of intervening in the gut ecosystem that have emerged. In May 2021, the International Scientific Association of Probiotics and Prebiotics (ISAPP) published a consensus statement on postbiotics, stating that postbiotics refer to “preparations of inanimate microorganisms and/or their components that confers a health benefit on the host” [14]. The efficacy of postbiotics is based on microbial metabolites, proteins, lipids, carbohydrates, vitamins, organic acids, cell wall components, or other complex molecules produced in the fermentation matrix. At this point the viability of the microorganism is no longer an important criterion. Therefore, postbiotics have advantages that are unmatched by probiotics, including a well-defined chemical structure, safe dosing parameters, longer shelf life, and better absorption, metabolism, and organismal distribution. The use of postbiotics can obtain probiotic-like efficacy while avoiding the problems of low bioavailability of bacteria, unstable effects, easy transmission of drug-resistant genes, easily cause microecological imbalance, increased microbial translocation, opportunistic infection or enhanced inflammatory response, etc. It is considered a better and safer strategy and will be a new direction for future research in the field of probiotics [15,16]. At present, research on postbiotic function is still in its infancy, and further studies would be needed to unveil other beneficial effects and clarify the probiotic mechanisms of postbiotics. Furthermore, there is a lack of relevant studies directly comparing the clinical benefits of postbiotics and probiotics to support their application.

The aim of this work was to characterize the therapeutic effects of *S. boulardii* and its postbiotics on a DSS-induced colitis model in mice, focusing on the alteration of the intestinal microbiota and its contribution to colitis-related parameters. It also explores the role of *S. boulardii* and its postbiotics in the intestinal microbial ecosystem to reveal the potential mechanisms of its intestinal anti-inflammatory activity, and to initially compare the effectiveness between postbiotics and probiotics. The highlight is that this study visually compares the differences in the therapeutic effects of the *S. boulardii* and its postbiotics to provide a theoretical basis and practical guidance for the development of effective modalities to alleviate colitis and related inflammatory bowel diseases.

## 2. Experimental Section

### 2.1. Animals

Male specific pathogen-free (SPF) C57BL/6 J mice (age 6–8 weeks, weight 18–22 g) were purchased from Beijing Vital River Laboratory Animal Technology Co., Ltd. (SCXK (Beijing, China) 2021-0006). All mice were housed in a standard SPF environment in the animal center (SYXK (Jing) 2020-0052). Five mice were maintained in each individually ventilated cage (temperature, 22 ± 2 °C; relatively humidity, 40–70%; standard 12 h/12 h light/dark cycle).

### 2.2. Chemicals and Reagents

Yeast Extract Peptone Dextrose Medium was purchased from Qingdao Hi-Tech Park Haibo Biotechnology Dextran sulfate sodium (DSS, *w*/*v*. molecular mass 36–50 kDa) was purchased from Yisheng Biotechnology Co., Ltd. (Shanghai, China). RNA Extraction Kit (*TransZol*™ Up Plus RNA Kit) and cDNA Synthesis Kit (*TransScript*^®^ One-Step gDNA Removal and cDNA Synthesis SuperMix) were purchased from Beijing TransGen Biotechnology Co., Ltd. (Beijing, China). SuperReal Color Fluorescence Quantitative Premix Reagent was purchased from Tiangen Biochemical Technology Co., Ltd. (Beijing, China).

### 2.3. Probiotics and Postbiotics

*Saccharomyces boulardii* was isolated from Angel Fubon Colibri Ken feed additive, strain conservation number CCTCC NO.M 2012116.

Freeze-dried *S. boulardii* postbiotic: YPD liquid culture was sterilized at 121 °C for 15 min, and after cooling to 55–60 °C, the *S. boulardii* strain was inoculated and incubated at 30 °C with shaking at 200 rpm until the stable stage. The fermentation broth was divided into portions, and after the pre-freezing was completed, the powder was freeze-dried by a freeze-dryer, and the lyophilized postbiotic sample was obtained.

Spray-dried *S. boulardii* postbiotic: YPD liquid culture was sterilized at 121 °C for 15 min, and after cooling to 55–60 °C, the *S. boulardii* strain was inoculated and incubated at 30 °C with shaking at 200 rpm until the stable stage. The spray-drying conditions were set as follows: the inlet temperature was set at 135–140 °C, the outlet air temperature was between 48 °C and 50 °C, the inlet flow rate and the drying air flow rate were 7.5 mL/min and 32.5 m^3^/h, respectively. The powder obtained was the spray-dried postbiotic sample.

### 2.4. Animal Experiment

After fifty C57BL/6J male mice were acclimatized and fed in the experimental environment for 1 week, the mice were divided into five groups (*n* = 10): Control, DSS, BLD (*S. boulardii*), D-BLD (Freeze-dried *S. boulardii* postbiotic), and P-BLD (Spray-dried *S. boulardii* postbiotic) groups. Mice in the DSS, BLD, D-BLD, and P-BLD groups were administered with DSS (2%) in drinking water daily for 7 days. The 2% DSS solution was changed once a day for 7 days. The mice were weighed daily and their disease activity index (DAI) was assessed according to the scoring criteria as shown in Appendix A. Then, probiotics and postbiotics were administered to mice in BLD, D-BLD, and P-BLD groups by oral gavage from day 7 for 7 consecutive days. While the mice in Control and DSS groups were gavaged with 200 μL of sterile phosphate buffered saline (PBS) solution. Both freeze-dried and spray-dried *S. boulardii* postbiotics were dissolved in PBS solution (Figure 1). From day 7, all animals had unrestricted access to food and water. After the last feeding, mice were fasted without water for 12 h and were sacrificed on day 15. Before the mice were sacrificed by cervical dislocation, the blood samples were taken from the posterior orbital venous plexus. The serum was obtained and then stored at −80 °C. The organs were dissected immediately. The liver, kidney, and spleen were weighed to calculate organ indices. Organ index (organ index of spleen, liver, and kidney) = organ weight/body weight × 100%. The length of colon was measured. A segment of the colon was flushed with sterile water and then fixed in 4% paraformaldehyde for subsequent histopathological analysis. The colon and cecal contents were snap-frozen in liquid nitrogen instantly, and then kept at −80 °C for further investigation.

### 2.5. Histopathological Analysis

The distal colon (2 cm from the anus) was taken and fixed in 4% paraformaldehyde for histopathological assessment. Paraformaldehyde-fixed colon tissues embedded in paraffin and sectioned into 4 µm thick sections. Paraffin sections were dewaxed and stained with hematoxylin and eosin. A SUNNY EX20 biological microscope (Ningbo Sunny Instruments Co., Ltd., Ningbo, China) was used to examine the sections.

### 2.6. Measurement of Serum Inflammatory Indicators

The blood samples were placed at rest for 2 h and then centrifuged once at 4 °C (3500 r/min, 15 min) to collect the serum. The colonic concentration of inflammatory factors including tumor necrosis factor (TNF)-α, interleukin (IL)-1β, IL-6, and IL-10 were measured by corresponding ELISA kits according to the instructions of manufacturer.

### 2.7. Quantitative Real-Time Polymerase Chain Reaction Analysis for mRNA Expression

Total RNA was extracted from the colon using TransZolTM Up Plus RNA Kit according to the manufacturer’s protocols. The extracted RNA was reverse transcribed to complementary DNA (cDNA) using a reverse transcriptase kit. The cDNA was analyzed by quantitative real-time PCR (qRT-PCR) using SuperReal PreMix Plus (SYBR Green) (Tiangen Biochemical Technology Co., Ltd., Beijing, China). Each reaction was subjected to the following cycling conditions: a pre-cycling stage at 95 °C for 15 min, followed by 40 cycles at 95 °C for 15 s, 60 °C for 20 s, and 72 °C for 25 s. The 2^−ΔΔCt^ method was used to determine the messenger RNA (mRNA) expression levels of cytokines and tight junction protein in colon tissue relative to the expression of GAPDH. The expression levels of pro-inflammatory cytokines (IL-1β, IL-6, and TNF-α), anti-inflammatory cytokines (IL-10), and tight junction protein (occludin) were evaluated. The sequences of the primers used in this study are provided in Table 1.

### 2.8. Gut Microbiota Analysis

Microbial genomic DNA was extracted and the bacterial V3–V4 regions of the 16 S rRNA were amplified and sequenced on a Nova-Seq platform (Illumina, San Diego, CA, USA). The raw data were filtered and analyzed by QIIME (v1.9.1). UPARSE v7.0.1001 was used to cluster the OTUs at an identity threshold of 97%. NMDS plots were analyzed using PAST v2.17 based on Bray–Curtis distance. Heat maps were drawn using HemI software (v1.0.3.7). Determinations of alpha and beta diversities were also conducted in QIIME (v1.9.1). The Majorbio Cloud Platform was used to create principal coordinates analysis (PCoA) visualizations (www.majorbio.com, accessed on 1 December 2022). In addition, this study used the Majorbio cloud platform (www.majorbio.com, accessed on 1 December 2022) to undertake a Bray–Curtis distance-based analysis of similarity (ANOSIM). Following the linear discriminant analysis (LDA), the LDA effect size (LEfSe) analysis was conducted to identify the differential bacterial taxa from the level of phylum to genus with two filters (*p* < 0.05 and LDA score > 2). PICRUSt was used to predict the functional profiles of microbial communities, and STAMP (version 2.1.3) was used to evaluate statistically significant differences (http://kiwi.cs.dal.ca/Software/STAMP, accessed on 1 February 2023).

### 2.9. Statistical Analysis

All data were expressed as means ± SEM and analyzed using GraphPad Prism 9.0 program (GraphPad Software, San Diego, CA, USA). All data differences were analyzed by one-way analysis of variance (ANOVA), followed by Tukey’s test, and *p* < 0.05 was considered as statistically significant.

## 3. Results

### 3.1. Both S. boulardii and Its Postbiotics Can Alleviate the Symptoms of DSS-Induced Colitis in Mice

By constructing a model of DSS-induced colitis, it was possible to detect and compare the beneficial effects of *S. boulardii* and its postbiotic application. The survival curves showed that *S. boulardii* and its postbiotics treatment temporarily delayed the death of the mice, which showed higher survival rates (Figure 2A). Weight loss, a typical feature of colitis, was significantly lower in mice in the DSS group compared with the Control group (Figure 2B, *p* < 0.01), the intervention reversed the weight loss, with the BLD and P-BLD groups (*p* < 0.01) showing better weight recovery than the D-BLD group (*p* < 0.05). The liver, kidney, and spleen indices of the mice in each group were examined, which can reflect the biological function of the organ to some extent. Among them, the spleen, as an important peripheral immune response organ, is capable of producing antibodies and other active substances, and is stimulated by immune activation in DSS-induced colitis thus causing enlargement. The results showed that the spleen index increased in the DSS group compared with the Control group, with a significant difference (*p* < 0.05). There were no significant changes for liver and kidney indices among all the groups (Figure 2C,D).

### 3.2. Both S. boulardii and Its Postbiotics Alleviated DSS-Induced Colonic Shortening and Pathological Changes in Mice

Shortening of colon length is one of the typical features of DSS-induced colitis. The colon length of mice in the DSS group was significantly shorter than that in the Control group (*p* < 0.01), and the colon length of mice in the three groups with *S. boulardii* and postbiotics intervention basically returned to normal, with no significant difference from that in the Control group (*p* > 0.05) as shown in Figure 3A,B. The results of Hematoxylin-eosin staining of colon sections are shown in Figure 3C. Under the light microscope, the Control group had normal colonic tissue structure, while the DSS group had disordered colonic tissue structure, incomplete colonic mucosa, irregular arrangement, damage or disappearance of glands and glandular lumen, and structural abnormalities of the crypt. Compared with the DSS group, *S. boulardii* and its postbiotics intervention could alleviate the damage of intestinal mucosa and crypt, inflammatory cell infiltration caused by DSS, and improve the histopathological changes in the colon caused by DSS. Among them, the degree of lesion reduction was more obvious in the postbiotic intervention group, which showed that the mucosa was basically intact, inflammation was less severe, and the crypt was basically intact and close to normal colonic tissue. Based on the above results, it can be concluded that both *S. boulardii* and its postbiotics can alleviate DSS-induced shortening of colonic length and lesions of colonic structural damage. The postbiotics’ effect may be superior to that of *S. boulardii*, but more evidence is needed for support.

### 3.3. Both S. boulardii and Its Postbiotics Regulated Serum Cytokine Levels in DSS-Induced Colitis Mice

The increase of pro-inflammatory factors and decrease of anti-inflammatory factors are typical features of colitis, which can reflect the development process and severity of colitis. The effect of *S. boulardii* and its postbiotics on serum inflammatory factors in mice with DSS-induced colitis was detected using an ELISA kit, and the results are shown in Figure 4. The levels of IL-6 and TNF-α in serum were significantly increased in the DSS group compared with the Control group (*p* < 0.001). Compared with DSS, the levels of TNF-α (*p* < 0.01) and IL-6 were significantly lower (*p* < 0.001) after the intervention of *S. boulardii* (i.e., BLD group). The intervention effect was more obvious in D-BLD and P-BLD groups, which showed a significant decrease in IL-6 and TNF-α levels (*p* < 0.001) and IL-1β levels (*p* < 0.05). There was no significant difference in IL-10 levels between the groups. The results showed that both *S. boulardii* and its postbiotics were able to alleviate the symptoms of colitis caused by DSS by reducing the expression of pro-inflammatory factors, and the intervention effects of postbiotics were better than those of *S. boulardii*. In summary, this shows that postbiotics may have stronger immunomodulatory effects.

### 3.4. Both S. boulardii and Its Postbiotics Modulated the Expression Levels of Inflammatory Factors and Tight Junction Proteins in the Colons of Mice with DSS-Induced Colitis

To further test the immunomodulatory effects of *S. boulardii* and its postbiotics, the expression of mRNA levels of the colonic inflammatory factors IL-1β, IL-6, IL-10, and TNF-α in each group of mice was examined using RT-qPCR in this study. As shown in Figure 5A, the gene expression levels of IL-1β and TNF-α were significantly higher (*p* < 0.001) and the level of IL-10 was significantly lower (*p* < 0.01) in the colon of the DSS group compared with the Control group. Compared with DSS, the colonic IL-1β levels were significantly lower and IL-10 levels were significantly higher after intervention with *Saccharomyces boulardii* and spray-dried postbiotics (BLD and P-BLD groups) (*p* < 0.001). The most significant intervention effect was observed in the D-BLD group, which showed a significant decrease in IL-1β and TNF-α levels and a significant increase in IL-10 levels (*p* < 0.001). There was no significant difference in IL-6 levels between the groups. Remarkably, it was observed that the intervention with *S. boulardii* and P-BLD did not decrease the expression of TNF-α and its level was even equal to that of the DSS group. In summary, both *S. boulardii* and its postbiotics can alleviate the inflammatory response of colon caused by DSS, among which the effect of freeze-dried postbiotics is the most obvious.

Disruption of the intestinal barrier is one of the characteristics of the pathogenesis of patients with ulcerative colitis. To investigate the effect of *S. boulardii* and its postbiotic elements on the intestinal barrier in mice, the mRNA expression level of the tight junction protein occludin in the colon of mice was analyzed, as shown in Figure 5B. Compared with the Control group, the expression level of occludin in the DSS group was reduced significantly (*p* < 0.05); it was slightly increased after the intervention of *S. boulardii* and postbiotics, but with no significant difference between the groups (*p* > 0.05).

### 3.5. Both S. boulardii and Its Postbiotics Modified Gut Microbiota in DSS-Induced Colitis Mice

Intestinal microorganisms are crucial for the development of inflammatory bowel disease, and improving colitis by regulating intestinal flora is a current research hotspot. The sequences of the genes from the V3-V4 regions of the intestinal contents of mice in each experimental group were sequenced with high throughput to detect changes in the intestinal flora.

#### 3.5.1. Gut Microbiota Overall Structure Was Altered by Both *S. boulardii* and Its Postbiotics

The overall structure of the intestinal flora was evaluated by diversity analysis of single samples (α diversity) and diversity analysis between samples (β diversity). The Chao index as well as the Simpson index reflected the α diversity of the intestinal microbial populations, and Figure 6A,B shows that based on the analysis of OTU level, the Chao index of the intestinal flora of DSS-induced colitis mice and the Simpson index were not statistically different, indicating that DSS treatment, *S. boulardii*, and its postbiotic intervention did not significantly alter the intestinal microbial community richness and homogeneity in colitis mice.

The Venn diagram provides a visualization of the number of OTUs common and specific to multiple sample groups. The results are shown in Figure 6C. The Control, DSS, BLD, D-BLD, and P-BLD groups each had 503, 362, 292, 311, and 351 OTUs, and the number of endemic OTUs were 24, 33, 4, 11, and 15, respectively. Based on OTU abundance, principal component analysis (PCA) was used to determine overall differences in microbial communities among groups. The PLS-DA scale directly reflected differences between and within groups. PCoA and NMDS analyses were used to assess differences in microbiota structure. The results showed that the samples showed significant clustering, with significant differences (*p* < 0.01) in the structure of the flora between groups. The samples of the DSS group were far from the Control group, and the intervention of *S. boulardii* and its postbiotic elements changed the intestinal flora structure of mice back toward the Control group. All these results showed that the intervention of *S. boulardii* and its postbiotics would influence the structure of gut microorganisms.

#### 3.5.2. Gut Microbiota Composition Was Regulated by Both *S. boulardii* and Its Postbiotics

The composition of the intestinal microbial community and its structure play a crucial role in maintaining intestinal microbial homeostasis. Figure 7A–C analyzes the relative abundance of intestinal microorganisms in each experimental group at both the phylum and genus levels, with the Control group flora reflecting the composition of the normal mouse intestinal flora, to determine the differences in community composition between groups. As shown in Figure 7A, at the gate level, *Bacteroidota* (70.81%), *Firmicutes* (17.87%), *Actinobacteriota* (9.22%), and *Patescibacteria* (0.95%) were the four major groups of bacteria in the intestine of normal mice. Compared with the Control group, the relative abundance of *Bacteroidota* (64.84%) and *Actinobacteriota* (2.07%) decreased and the relative abundance of *Firmicutes* (27.46%) and *Patescibacteria* (3.72%) increased in the mice of the DSS group. Both *S. boulardii* and postbiotics interventions reversed these changes and showed a microbial community structure more similar to that of the Control group.

As shown in Figure 7B, at the genus level, norank_f__*Muribaculaceae* (64.70%), *Ileibacterium* (9.25%), *Bifidobacterium* (6.06%), *Alloprevotella* (4.41%), and *Dubosiella* (3.85%) were the main components of the intestinal flora genus level in Control group mice, which are key genera for maintaining intestinal microecological stability. Community structure changed after DSS induction, when norank_f__*Muribaculaceae* (37.96%), *Prevotellaceae*_UCG-001 (8.15%), *Alistipes* (7.78%), and *Lactobacillus* (7.40%) became the major components of their flora, and the relative abundance of all four major groups of the Control group was significantly reduced. *S. boulardii* and postbiotics could improve the DSS-induced intestinal flora dysbiosis and make the community composition closer to the normal level. The intervention effect was obvious, as follows: the relative abundance of norank_f__*Muribaculaceae* was adjusted upward from 37.96% to 69.10%, 62.68%, and 63.15%, respectively, for the BLD, D-BLD, and P-BLD groups, which were close to the normal level (64.70%). However, there were differences in community composition among the groups, with *Dubosiella* (10.27%), *Dubosiella* (4.83%), and *Alistipes* (5.79%) being the second most abundant groups in relative abundance in the BLD, D-BLD, and P-BLD groups, respectively. The D-BLD group had better community regulation than the other groups and showed a community composition closer to that of the Control group.

In order to clarify the differences among groups, we used linear discriminant analysis (LDA) of the effect amount (LEfSe) to determine the biomarkers with rich differences between groups. As shown in Figure 7C,D, each group is rich in different microbial communities, and species with significant differences are expressed by an LDA score greater than 2, which reflects the impact of species with significant differences in components. The results showed that gut microbiota in the mice of the DSS group was increased dramatically in Clostridia class, RF39, and Mycoplasma order. In addition, the DSS group had an increase in the relative abundance of *Erysipelatoclostridiaceae*, UCG-010, and *Mycoplasmataceae* at family level, and *Lachnonospiraceae*_NK4A136_group and *Erysipelatoclostridium* at genus level. Compared to the Control group, there were more diverse changes in the structure of the DSS colonic microbiota. The BLD group significantly increased the abundance of *Turcibacter* at the genus level. While the P-BLD group significantly increased the abundance of *Oscillospirales* and *Christensenellales* at the order level, *Ruminococcaceae*, *Christensenellaceae*, and *Defluviitaleaceae* at the family level, and *Turicibacter* and *Defluviitaleaceae*_UCG-011 at the genus level. The results suggested that *S. boulardii* and postbiotics may regulate the gut microbiota to achieve a new balance by restoring the abundance of dominant bacteria.

#### 3.5.3. Functional Profile of the Gut Microbiome Was Changed by Both *S. boulardii* and Its Postbiotics

In order to understand the potential function of colonic microbiota, we used the PICRUSt2 method to predict the KEGG function spectrum of intestinal microbiota based on 16S rRNA gene amplification sequence. Figure 8 shows the first 20 KEGG pathways between each group. After DSS induction, probiotics and postbiotics interventions, many pathways have changed significantly. The intestinal microbiota was mainly involved in multiple pathways such as global and overview maps, carbohydrate metabolism, amino acid metabolism, energy metabolism, and metabolism of cofactors and vitamins. The cardiovascular disease pathway decreased in the DSS group, the D-BLD group, and the P-BLD group compared to the Control group. With regard to drug resistance, the antimicrobial pathway increased in the DSS group compared with the BLD group and the P-BLD group. The administration of *S. boulardii* intervention reduced the decline in development and regeneration, immune disease, and substance dependence. Furthermore, D-BLD caused an increase in the development and regeneration pathway.

## 4. Discussion

Ulcerative colitis is a recurrent and persistent chronic inflammatory bowel disease with increasing incidence. It is a complex disease involving host, microorganisms, and other environmental factors, and its pathogenesis is still unclear, with strong drug dependence and more side effects. Thus, exploring effective nutritional intervention therapies with low toxic side effects has become a hot concern. Probiotics have been shown to improve the pathological signs of UC. The new concept of postbiotics has been proposed as a safe and effective intervention with great promise. However, very few studies have compared the actual effects of postbiotics and probiotics. The DSS-induced experimental colitis model is a well-established model that can be used to understand the pathogenesis of UC [17]. With the DSS model, we investigated for the first time the alleviative effect and mechanism of action of the probiotic *S. boulardii* and its postbiotics, with a view to its future application in UC treatment. Our results showed that both *S. boulardii* and its postbiotic elements could effectively alleviate weight loss, reduce colonic tissue damage, regulate the balance between pro/anti-inflammatory cytokines in serum and colon, promote the expression of colonic tight junction protein, and regulate the stability of intestinal microecology in mice, and these factors together alleviated the disease of DSS-induced colitis. Additionally, the combined therapeutic effects of the postbiotics were better than those of *S. boulardii*.

Firstly, we measured some macroscopic indicators, which were similar to the symptoms usually observed in some previous animal studies (Appendix A) [18,19,20,21,22,23,24,25] and human UC [26]. DSS-induced colitis mice showed weight loss, shortened colon length, and the presence of severe histological lesions, indicating that a DSS-induced colitis mouse model was successfully established in this study. In contrast, both the administration of *S. boulardii* and the postbiotics intervention were able to affect the survival rate, weight loss, colonic lesions, and length shortening in mice with colitis, indicating that it showed a healing effect on colitis by potentially reversing the above parameters. However, the results showed that there was no significant difference between the intervention group and the Control and DSS groups, except for the body weight, which was significantly different from the DSS group, which was somewhat different from the results of previous studies (Appendix A). It may be due to the different substances used, referring to different probiotic strains and postbiotic preparations, and different experimental designs such as experimental period and DSS dosage, which caused some of the results to be different.

The cytokine responses characterizing the inflammatory bowel diseases (IBDs) are the key pathophysiological factors controlling the initiation, evolution, and eventual resolution of these forms of inflammation [27]. Clinical studies have found that pro-inflammatory cytokines are highly correlated with the severity of colitis in patients with UC [14]. Studies have shown that fresh colonic mucus biopsies from UC patients show elevated mRNA expression levels of IL-1β, TNF-α, and IL-6 [28]. This is similar to the results we obtained, where serum TNF-α and IL-6 levels and mRNA expression levels of colonic TNF-α and IL-1β were significantly higher in the DSS group of mice than in the Control group. Studies have shown that probiotics have therapeutic effects on controlling inflammation and maintaining UC carcinogenesis in remission [29]. For example, *S. boulardii* has been shown to inhibit TNF-α and IL-6 levels as well as other pro-inflammatory cytokines such as IL-1β and IL-8 R, demonstrating that *S. boulardii* can reduce colonic inflammation and regulate inflammatory gene expression [30]. This was also confirmed by our study, in which *S. boulardii* significantly reduced DSS-induced serum TNF-α, IL-6, and colonic IL-1β mRNA expression levels in mice. In contrast, the two postbiotics showed a superior ability to intervene, as evidenced by their ability to significantly reduce DSS-induced serum TNF-α, IL-6, and IL-1β levels, as well as colonic IL-1β mRNA expression. D-BLD also significantly reduced colonic TNF-α mRNA expression. These are our new findings after comparison, suggesting that postbiotics may have a stronger immunomodulatory ability than probiotics, and that there are differences between postbiotics, which deserve more in-depth study, such as their specific mechanisms.

IL10 is an immunosuppressive cytokine produced by B cells, T cells, macrophages, and some non-hematopoietic cells upon stimulation [31]. IL-10 has a broad effect in immunoregulation and host defense, as it affects both the innate and adaptive immune systems [32]. We also examined the levels of this conventional anti-inflammatory factor, and although there was no significant difference in serum levels between the groups, IL-10 mRNA expression levels in the colon were inhibited by DSS (*p* < 0.01). This was improved by the intervention of *S. boulardii* and postbiotics, and the postbiotics were superior to the probiotics. The above results suggested that both *S. boulardii* and its postbiotics have immunomodulatory abilities and can improve the balance of pro/anti-inflammatory factors in the serum and colon of DSS-induced mice, while the postbiotics have a stronger immunomodulatory ability compared with probiotics.

It is well documented that pro-inflammatory cytokines are involved in the pathogenesis of colitis, and in addition to their contribution to intestinal mucosal inflammation, they are also associated with the intestinal epithelial barrier. For example, cytokines of the IL-10 family are essential for maintaining the integrity and homeostasis of the tissue epithelial barrier, suggesting that they can promote the innate immune response of the tissue epithelium, limit the damage caused by viral and bacterial infections, and promote the healing process of damaged tissues caused by inflammation [33,34,35]. Intestinal epithelial cells (IECs) maintain a fundamental immunoregulatory function that influences the development and homeostasis of mucosal immune cells. The intestinal epithelial barrier is extremely important to protect the host from exogenous pathogens [36]. The association between increased bacterial translocation and risk of developing inflammatory bowel disease (IBD) suggests a central role for dysregulated epithelial barrier function in either the etiology or the pathology of intestinal inflammation and IBD [37]. Damage in the intestinal epithelium, as observed during UC, seriously affects barrier function and results in malabsorption, chronic inflammation, and diarrhea [38]. DSS-induced colitis is manifested by significant damage to intestinal epithelial cells and loss of intestinal barrier function. Since the expression of tight junction protein is a major determinant of intestinal barrier function, we investigated the mRNA expression levels of colonic occludin in various groups of mice. The upregulation of tight junction protein expression in the intervention group provides an alternative explanation for the anti-colitis effect of *S. boulardii* and postbiotics. In conclusion, in the present study, we found that *S. boulardii* and postbiotics may have positive effects on tight junctions. Therefore, more future investigations are warranted to determine the effects of *S. boulardii* and postbiotics on tight junctions, in order to provide viable basis for clinical treatments for acute colitis.

In addition to intestinal permeability, the homeostasis of the gut microbiota is another important factor of concern. The symbiotic gastrointestinal microbiota provides a variety of beneficial services to the healthy host, including maintenance of immune homeostasis, regulation of gastrointestinal development, and enhanced metabolic capacity [39]. To better understand the possible role of gut flora in UC, we examined the changes in gut flora in various groups of mice and compared them. Some previous studies have shown that inflammatory bowel disease can reduce the diversity of gut microbiota, change the composition of gut microbiota and lead to the destruction of the intestinal microecology [40,41,42,43]. Our results showed that, although there was no significant difference in alpha diversity between the DSS group compared to the postbiotic or probiotic groups, the intervention modulated the beta diversity of the fecal microbiota in mice with colitis. Both *S. boulardii* and postbiotics interventions significantly altered the structure and composition of the intestinal microflora and reshaped the intestinal microbial changes caused by DSS.

To understand the difference of microbial composition among different groups, we analyzed microbial composition at the phylum level and the genus level. The DSS-induced colitis model involves the actions of various microbes, such as *Firmicutes*, *Bacteroidetes*, and *Patescibacteria*. *Bacteroides* and *Firmicutes* are the main dominant phyla in gut microbiota, which are involved in the host’s energy homeostasis regulation [44]. Accumulating evidence demonstrated that *Firmicutes* and *Bacteroidota* played a critical role in UC development [45]. For example, one study showed that *Bacteroidota* displayed negative correlations with the metrics of UC activity, so the loss of these species is suggested to result from UC exacerbation [46]. This could be because *Bacteroidetes* species adhering to the mucosal surface may be unable to inhabit the niche of the extensively damaged mucosa without sufficient mucin production in highly severe UC [47]. *Firmicutes* play a significant role in gut homeostasis through the production of metabolites, which enhances the gastrointestinal barrier and mucosal immune functions [48]. Herein, based on metagenomics sequencing, the abundance of *Bacteroidota* decreased in the DSS group, while that of *Firmicutes* increased, but both phyla showed opposite trends after *S. boulardii* and postbiotics treatments. This result is similar to previous reports [49], some studies have also reached the opposite results [50], but we confirmed that *S. boulardii* and postbiotics interventions can remodel the DSS-induced intestinal microbiota on the phylum level to more closely resemble normal mice.

At the genus level, *Muribaculaceae* which belong to *Bacteroidota*, was the major colonic microbiota in each treatment group. *Muribaculaceae* is a major microbiota occurring in the intestinal tract of various animals and has been identified as a fermenter capable of producing succinate, acetate, and propionate [51]. Our results showed that *Muribaculaceae* was significantly reduced in the DSS group which was reversed by *S. boulardii* and postbiotics interventions. In contrast to *Muribaculaceae*, our results showed that abundance of the *Lachnospiraceae*_NK4A136_group in the DSS group was relatively higher than that of the Control group. Previously, *Lachnospiraceae* exhibited higher abundance in both DSS-treated mice and IBD patients [52]. The *Lachnospiraceae*_NK4A136_group positively correlated with the pathological characteristics of chemically induced mouse models of colitis, including DSS [53]. In the model of chronic colitis, we show that imbalance of *Muribaculaceae* and *Lachnospiraceae* caused by DSS were ameliorated in the mice orally gavaged with *S. boulardii* and postbiotics interventions. Similar to our findings, the probiotic yeast BR14 [54] and *L. plantarum* JS19 [55] rebalanced the composition of gut microbiota in the mice with DSS-induced colitis by increasing the abundance of *Muribaculaceae* and decreasing the abundance of *Lachnospiraceae*.

We also used imputed relative abundances of KEGG pathways in each sample to predict changes in metabolic function in microbiomes. For the imputed relative abundances of KEGG pathways in different groups, the greatest statistical differences were observed for the development and regeneration, cardiovascular disease, and immune disease. Although the analysis shows fewer KEGG difference paths between groups, these results indicated that *S. boulardii* and postbiotics treatments significantly affected these pathways in this study. It is important to note that, while PICRUSt2 is a relatively robust metagenomic prediction tool, it is entirely dependent on the quality of the reference database used and is therefore incapable of accounting for strain variations present in a given community [56,57]. At the very least, these results warrant further investigation into the role of the gut microbiota in DSS-induced colitis.

A recent study showed that UC is indeed characterized by abnormal mucosal immune response, but microbial factors (changes in the composition of intestinal microbial flora) and epithelial cell abnormalities (abnormalities in the function of epithelial barrier) can promote this response [58]. Therefore, in general, our direct comparison of probiotic *S. boulardii* and its postbiotics showed that they have a certain degree of UC-related symptoms relief, but postbiotics have a stronger ability to alleviate UC inflammatory symptoms and regulate intestinal microbiota. However, we acknowledge that this result has certain limitations and still needs to be supported by more evidence, for the following reasons: 1. This study uses the model of acute colitis, and the treatment intervention time is short. In the future, it may be possible to obtain more obvious or accurate experimental results through a longer modeling and treatment cycle, such as using the model of chronic colitis, and extending the intervention cycle. 2. There is a lack of general criteria for the evaluation of results. For example, it is not yet possible to judge the difference between the therapeutic effects of epigenetic agents and conventional therapeutic drugs, such as mesalazine, to judge its therapeutic efficiency more accurately. 3. Although epigenetic agents are generally considered to be of low risk, existing studies cannot provide evidence to show the safety of *S. boulardii* postbiotics, and a large number of animal models and human clinical studies are still needed to verify this hypothesis. In conclusion, postbiotics as an extension direction of probiotics has great potential, such as how *Lactobacillus plantarum*-derived postbiotics ameliorate acute alcohol-induced liver injury [59]. Therefore, the multiple applications of postbiotics will be an effective supplement to probiotics and a driving force for the development of the total health industry [60].

## 5. Conclusions

In summary, the present study demonstrates that *S. boulardii* and its postbiotics can effectively alleviate DSS-induced colitis in mice by reducing the inflammatory response and maintaining intestinal homeostasis. Our findings highlight the differential function of probiotics and even different postbiotics in colitis remission, it would be of interest to further explore the exact mechanisms of action of probiotics and postbiotics. In the future, more precision research should be carried out on specific postbiotics prepared by different strains. To the best of our knowledge, this is the first study to systematically compare the performance of probiotic *S. boulardii* and its postbiotics derived from the same strain in an experimental colitis model. Our results highlight postbiotics as a promising next generation biotherapeutic for ulcerative colitis treatment.

## Figures and Tables

**Figure 1 nutrients-15-01484-f001:**
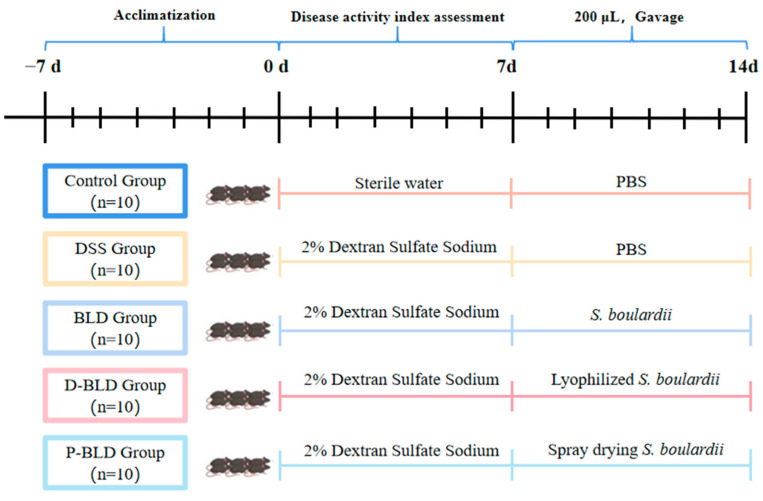
Experimental design.

**Figure 2 nutrients-15-01484-f002:**
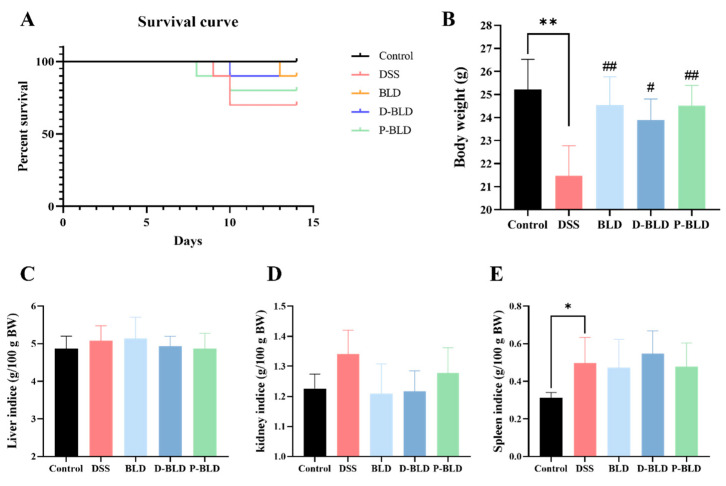
Effect of *S. boulardii* and its postbiotics on the survival curve, body weight, and organ index of mice with DSS-induced colitis. (**A**) Survival curve. (**B**) Body weight. (**C**) Liver index. (**D**) Kidney index. (**E**) Spleen index. Note: Data are expressed as mean ± SD. Comparison of significant differences between groups is indicated, where * *p* < 0.05, ** *p* < 0.01 compared with Control group; # *p* < 0.05, ## *p* < 0.01 compared with DSS group.

**Figure 3 nutrients-15-01484-f003:**
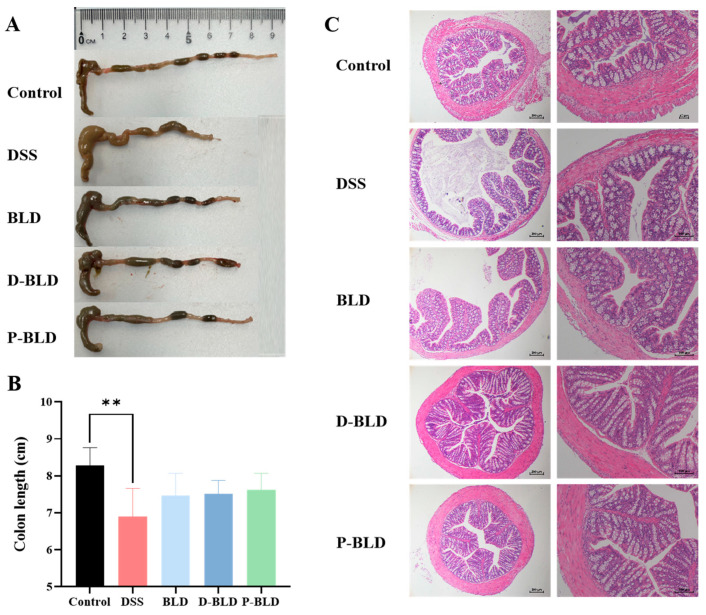
Effects of *S. boulardii* and its postbiotic elements on colon length and colon tissue structure in mice with DSS-induced colitis. (**A**) Apparent picture of colon tissue. (**B**) length of colon. (**C**) HE staining result of colon tissue. The first column of the scale bar is 200 μm, and the second column of the scale is 100 μm. Note: Data are expressed as mean ± SD. Comparison of significant differences between groups is indicated, where ** *p* < 0.01 compared with Control group.

**Figure 4 nutrients-15-01484-f004:**
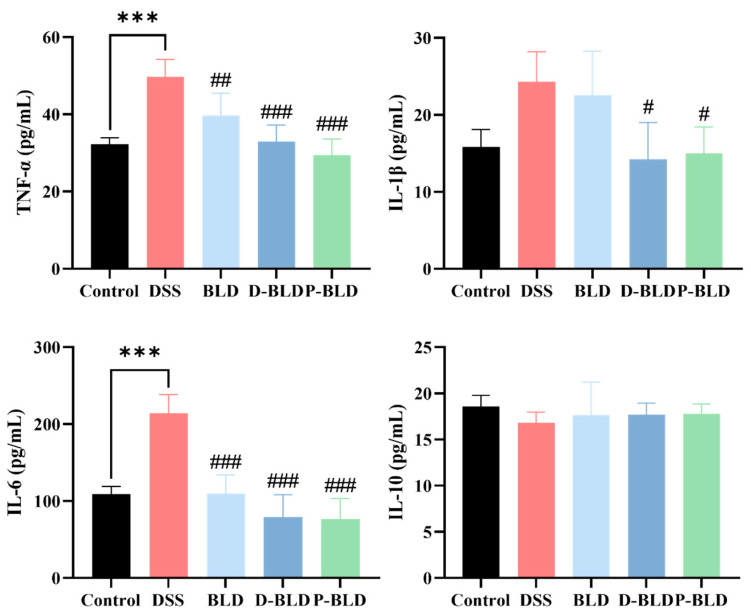
Effect of *S. boulardii* and its postbiotic elements on serum inflammatory factors in mice with DSS-induced colitis. Note: Data are expressed as mean ± SD. Comparison of significant differences between groups is indicated, where *** *p* < 0.001 compared with Control group; # *p* < 0.05, ## *p* < 0.01, ### *p* < 0.001 compared with DSS group.

**Figure 5 nutrients-15-01484-f005:**
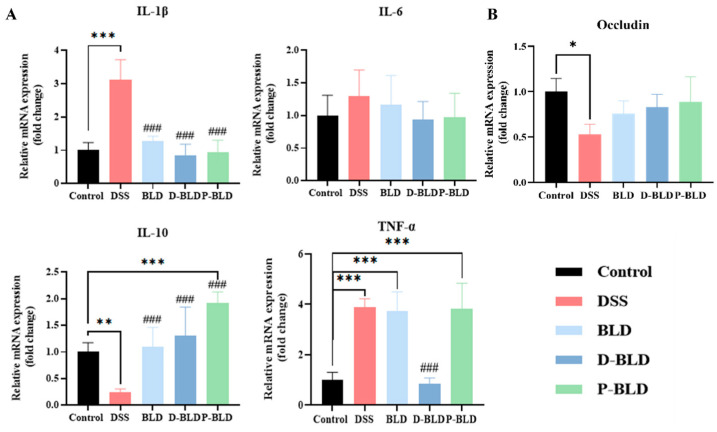
Effect of *S. boulardii* and its postbiotic elements on colonic inflammation and relative expression levels of mRNA of intestinal barrier-related genes in mice with DSS-induced colitis. (**A**) Inflammatory factors. (**B**) Tight junction proteins. Note: Data are expressed as mean ± SD. Comparison of significant differences between groups is indicated, where * *p* < 0.05, ** *p* < 0.01, *** *p* < 0.001 compared with Control group; ### *p*< 0.001 compared with DSS group.

**Figure 6 nutrients-15-01484-f006:**
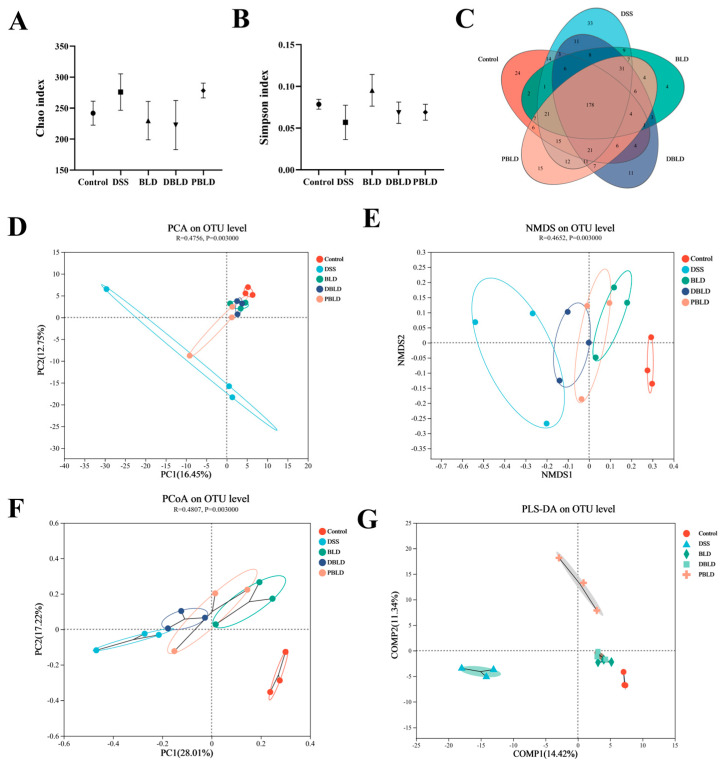
Analysis of intestinal microbial diversity. (**A**) Chao index. (**B**) Simpson index. (**C**) Venn diagram. (**D**) PCA analysis. (**E**) NMDS analysis. (**F**) PCoA analysis. (**G**) PLS–DA analysis. Note: The above gut microbial analyses were based on OTU levels.

**Figure 7 nutrients-15-01484-f007:**
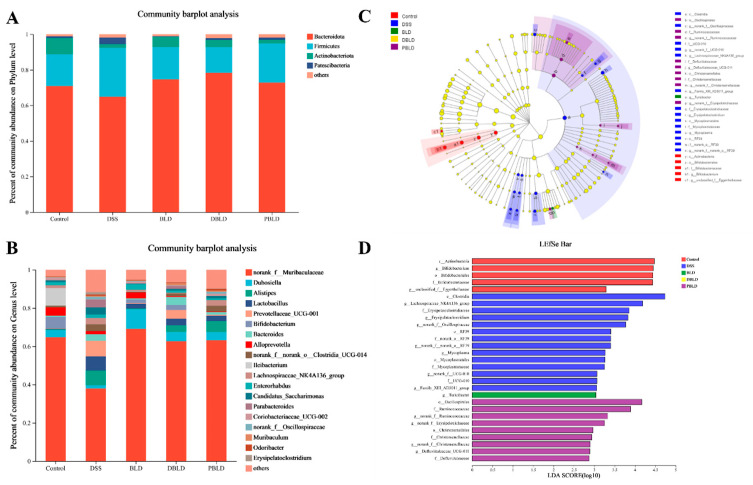
Analysis of the composition of the gut microbial community. (**A**) Composition of the gut flora at the portal level. (**B**) Composition of the gut flora at the genus level. (**C**) LEfSe Cladogram (the rings, from outer to inner, represent genus, family, order, class, and phylum). (**D**) Indicator bacteria with LDA scores of >2 in five groups.

**Figure 8 nutrients-15-01484-f008:**
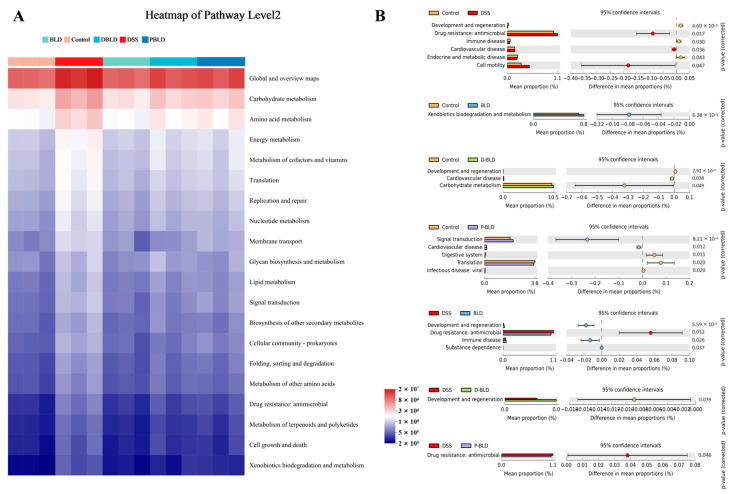
Predicted function of the fecal microbiome based on KEGG pathways. (**A**) Heatmap of Pathway Level2. (**B**) Important pathways associated between two groups. Statistically significant difference was performed using the unpaired two-tailed Welch’s *t*-test (*p* < 0.05) in STAMP.

**Table 1 nutrients-15-01484-t001:** Sequences of primers used for quantitative real-time PCR (qRT-PCR).

Target Gene	Nucleotide Sequence of Primer (5′ to 3′) Forward	Reverse
GAPDH	ATGGTGAAGGTCGGTGTGAA	TTTGCCGTGAGTGGAGTCAT
IL-1β	GTCGCTCAGGGTCACAAGAA	CCACACGTTGACAGCTAGGT
IL-6	GGAGCCCACCAAGAACGATA	GTCACCAGCATCAGTCCCAA
IL-10	AGAGAAGCATGGCCCAGAAA	ACACCTTGGTCTTGGAGCTT
TNF-α	AGATTCTTCCCTGAGGTGCA	ACCCCGGCCTTCCAAATAAA
Occludin	TTTCCTGCGGTGACTTCTCC	AAAACAGTGGTGGGGAACGT

## Data Availability

Not applicable.

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
