# Peer review of "Both Saccharomyces boulardii and Its Postbiotics Alleviate Dextran Sulfate Sodium-Induced Colitis in Mice, Association with Modulating Inflammation and Intestinal Microbiota"

_nutrients, 2023, doi:10.3390/nu15061484_

Round 1

Reviewer 1 Report

Xu et al. described the effect of S. boulardii and its postbiotics on the DSS-induced induced colitis using a mouse model. The authors showed that S. boulardii and its postbiotics reduce the symptoms of DSS-induced colitis, which were assessed by colon lengths and pathological changes. BLD-received groups also displayed the reduced colonic expressions pro-inflammatory cytokines at both protein and mRNA levels. Microbial genomic DNA sequencing revealed that S. boulardii and its postbiotics help maintain the homeostasis of intestinal microorganisms in mice with DSS-induced colitis. Overall, the authors robustly showed the effect of S. boulardii and its postbiotics on various markers of colitis. The study is novel in that it compares the effect of probiotics and postbiotics, particularly using microbial sequencing. I have a few minor concerns that need to be addressed:

1) In Fig 1C-E, what do the liver, kidney, and spleen indices exactly mean? It sounds like a vague term. if they mean tissue weights, I suggest the authors say liver weights, kidney weights, and spleen weights.

2) Line 208-211: I do not see any difference in the spleen index between control group and 3 BLD-received groups. But I see some levels of alleviation in the kidney index. Is this simply a typo?

3) Rather than stating “same below” (Figure 2 legend), please indicate the statistical analysis in the Figure 3-5 legends to help readers’ readability.

4) In line 235: avoid using “some”, but specify the alleviated symptoms.

5) In the abstract (line 26) and in the conclusions (line 563), I suggest the authors remove “enhancing intestinal barrier function”. The only data regarding the intestinal barrier function is Fig 5B, where BLD groups displayed non-statistically significant increase in occludin mRNA expression, which is not convincing to conclude so. Or the authors may want to include more data to support the conclusion in the revised manuscript.

6) On a related point to question 5, line 470- 478 in the discussion is not convincing. Again, the authors will need to adjust the manuscript unless additional data support the notion.

Author Response

Response to Reviewer 1

We gratefully thank the editor and all reviewers for their time spend making their constructive remarks and useful suggestions, which has significantly raised the quality of this manuscript and has enable us to improve the manuscript. Each suggested revision and comment, brought forward by the reviewers was accurately incorporated and considered. Below the comments of the reviewers are response point by point and the revisions are indicated.

We are very grateful to the Reviewer 1 for giving us very professional and specific suggestions to revise our manuscript. Point by point responses to reviewer 1 are as following: 

1) In Fig 1C-E, what do the liver, kidney, and spleen indices exactly mean? It sounds like a vague term. if they mean tissue weights, I suggest the authors say liver weights, kidney weights, and spleen weights.

Thank you for your comment. We are very sorry for the inconvenience in this manuscript they caused in your reading. The organ indice, also known as the relative weight of organs, is the ratio of the weight of an organ to its body weight in experimental animals, which can initially reflect the health condition of the animal. We have added the description of the organ indice in line 145-146 of the latest manuscript, Which will help readers understand more details. Thank you again for your valuable advice.

2)Line 208-211: I do not see any difference in the spleen index between control group and 3 BLD-received groups. But I see some levels of alleviation in the kidney index. Is this simply a typo?

Thank you so much for your careful check. The results on the spleen index are presented in line 213-214 of the latest manuscript, according to our data analysis, as we described and you saw “the spleen index increased in the DSS group compared with the Control group, with a significant difference (P<0.05), and the intake of S. boulardii and its postbiotics were able to partially alleviate the enlargement of the spleen caused by colitis, but there was no significant difference.” To avoid this confusion again, we have added "There were no significant changes for liver and kidney indices among all the groups (Figure 1C&D)." in line 214-216 of the latest manuscript. Thanks again for your comments.

3)Rather than stating “same below” (Figure 2 legend), please indicate the statistical analysis in the Figure 3-5 legends to help readers’

Thanks for your comment. We are sorry for any inconvenience this may cause. We have revised it in the manuscript.

4) In line 235: avoid using “some”, but specify the alleviated symptoms.

Thank you very much for your careful inspection and we have revised it in the manuscript. We have its expression as "shortening of colonic length and lesions of colonic structural damage", in the latest manuscript line 241-242.

5) In the abstract (line 26) and in the conclusions (line 563), I suggest the authors remove “enhancing intestinal barrier function”. The only data regarding the intestinal barrier function is Fig 5B, where BLD groups displayed non-statistically significant increase in occludin mRNA expression, which is not convincing to conclude so. Or the authors may want to include more data to support the conclusion in the revised manuscript.

We appreciate for your constructive comment. We quite agree with your suggestions and follow your suggest we have removed“enhancing intestinal barrier function”in the manuscript. We understand that the comments will help us improve the quality of our articles. We clearly recognize that more data of enhancing intestinal barrier function will further enhance the validity of the conclusion. Unfortunately, the results are still being explored. We will attach importance to this suggestion in the future writing process and thank you again for your professional comment.

6)On a related point to question 5, line 470- 478 in the discussion is not convincing. Again, the authors will need to adjust the manuscript unless additional data support the notion.

We would like to thank you for your thoughtful review of our manuscript. Although our results are not convincing enough, related studies suggest that the effect of postbiotic elements seems to be related to the intestinal barrier. We sincerely hope that through this article we can share with readers and arouse the attention of scholars on the function of postbiotics. Therefore, we have chosen to retain that part of the discussion and add suggestions for follow-up studies in line 497-499 of the latest manuscript. Thank you again for your constructive suggestions. We will make more efforts in this direction and hope to get your more advice future.

Reviewer 2 Report

In this study submitted to Nutrients, the authors assessed the protective effects of Saccharomyces boulardii and its freeze-dried and spray-dried postbiotics on dextran sulfate sodium-induced colitis in mice. The evaluation of Saccharomyces boulardii postbiotic effect is new.

The experimental design is good, no group is lacking. The choice of methods is good and comprehensive. All the parts of this paper are clear, well organized, with accurate references, sufficient details, no misinterpretation. The discussion is well written.

The authors revealed a better alleviation of colitis by postbiotic than live S boulardii. This is a promising finding because of the better safety of postbiotics compared to probiotics.

There are minor mistakes which require to be corrected.

Line 197-198. Please revise.

Lines 208-211 : “the intake of S. boulardii and its postbiotics were able to partially alleviate the enlargement of the spleen caused by colitis, but there was no significant difference. “

Actually, there is not even a trend. So this should be revised.

Line 228. The end of the sentence is lacking.

Line 251. Please avoid the repetition of “was significantly lower”

Line 254. Please revise: Il-6 should be replaced by IL1b.

Line 383. Please replace “antineoplastic” by “antimicrobial”

Author Response

Response to Reviewer 2

We gratefully thank the editor and all reviewers for their time spend making their constructive remarks and useful suggestions, which has significantly raised the quality of this manuscript and has enable us to improve the manuscript. Each suggested revision and comment, brought forward by the reviewers was accurately incorporated and considered. Below the comments of the reviewers are response point by point and the revisions are indicated.

We are very grateful to the Reviewer 2 for giving us very professional and specific suggestions to revise our manuscript. Point by point responses to reviewer 2 are as following: 

  1. Line 197-198. Please revise.Actually, there is not even a trend. So this should be revised.

We gratefully appreciate for your careful comment. We have revised it in the manuscript. (Line 204-205).

  1. Lines 208-211 : the intake of boulardiiand its postbiotics were able to partially alleviate the enlargement of the spleen caused by colitis, but there was no significant difference. Actually, there is not even a trend. So this should be revised.

Thank you for your careful consideration. We have revised it in the manuscript. 

  1. Line 228. The end of the sentence is lacking.

Thank you so much for your careful check. It's really our carelessness that caused the problem. We feel sorry for our carelessness. We have corrected it and we also feel great thanks for your point out. (Line 234).

  1. Line 251. Please avoid the repetition of “was significantly lower”

We are very sorry for inconvenience they caused in your reading. We have revised it in the manuscript. 

  1. Line 254. Please revise: Il-6 should be replaced by IL1b.

We feel sorry for our carelessness. We have corrected it and we also feel great thanks for your point out. 

  1. Line 383. Please replace “antineoplastic” by “antimicrobial”

We have carefully checked the manuscript and corrected this error accordingly. 
